# Microenvironment Molecular Profile Combining Glycation Adducts and Cytokines Patterns on Secretome of Short-term Blood-derived Cultures during Tumour Progression

**DOI:** 10.3390/ijms21134711

**Published:** 2020-07-01

**Authors:** Maria Laura Coluccio, Ivan Presta, Marta Greco, Rita Gervasi, Domenico La Torre, Maria Renne, Carlo Pietro Voci, Lorenzo Lunelli, Giuseppe Donato, Natalia Malara

**Affiliations:** 1University of Magna Graecia, 88100 Catanzaro, Italy; coluccio@unicz.it (M.L.C.); presta@unicz.it (I.P.); marta.greco@unicz.it (M.G.); dlatorre@unicz.it (D.L.T.); gdonato@unicz.it (G.D.); 2Mater Domini Hospital, 88100 Catanzaro, Italy; dr.ritagervasi@gmail.com (R.G.); mrenne@unicz.it (M.R.); voci.carlopietro@unicz.it (C.P.V.); 3Fondazione Bruno Kessler, 38123 Trento, Italy; lunelli@fbk.eu; 4CNR Institute of Biophysics, 38123 Trento, Italy

**Keywords:** secretome, cancer cells, liquid biopsy, blood-derived cultures, atomic force microscopy image, oxidation profile, methylglyoxal, personalized medicine

## Abstract

Cancer cells are known to secrete many bioactive factors acting both with paracrine and autocrine mechanisms by which they condition the surrounding microenvironment. At the same time, the intracytoplasmic metabolic activities microenvironment influences the profile of this secretion. It is well known that cancer cells exhibit prevalent glycolytic metabolism and a more oxidative atmosphere compared to their healthy counterparts; this metabolic phenotype promotes glycate adducts formation and secretion. Considering the exacerbation of metabolic changes during the cancer progression, it is suggestive to explore the potential correlation between the increasing rate of glycan adducts and the specific pattern of secreted cytokines in different phases of cancer disease. We analyzed the secretomes of blood-derived cancer cell cultures from cancer patients and healthy subjects. The relative glycate adducts content in cancer secretomes was higher in comparison to that of healthy samples. Moreover, the stratification based on different phases of cancer disease correlated with a specific cytokines panel. The results obtained open a new perspective of observation of the intricate relationship between metabolome and inflammation in cancer. By using the analysis of secretome combined with a standardized protocol of liquid biopsy, it would be possible to identify specific profiles of molecular markers useful to arrange alternative and personalized medicine strategies.

## 1. Introduction

The isolation of cancer cells for their profiling has always been a challenge, especially in the advanced stages of the disease. Metastatic cancer patients are often in precarious clinical conditions and unable to undergo further biopsy procedures. Nevertheless, to understand the illness it is necessary to gain more information above all from the molecular point of view, to portray the individual progression of the disease. When the information can be acquired from live and viable cancer cells, these data become very valuable to the patient’s evaluation both in the prognostic and therapeutic setting. Over the last few years, circulating tumour cells have become the subject of researches for their great potential in indicating cancer occurrence, but also in providing fundamental information about the physiology of cancer cells.

Actually, pre-clinical model systems, reflecting patient-specific clinical situations, are lacking. Much progress has currently been made in the study of circulating nucleic acids from tumour cells, but coupling appropriate therapies to the effects of genetic mutations is currently limited by an incomplete understanding of the interplay between the tumour cell and its microenvironment. Therefore, the development of patient-derived primary cancer cell cultures, using liquid biopsy, could represent a functional strategy for diagnostics and possibly a promising start point to develop specific assays for individual cancer patients [1].

Over the last few years, circulating tumour cells have become the subject of researchers for their great potential as a non-invasive diagnostic and prognostic biomarker also for their potential in providing functional and molecular data [2].

The main problem in their usage is the limited availability due to a relative scarce amount. Many techniques for their isolation have been developed and are continuously improving. In our laboratory, from 2008 an isolation protocol is used for increasing the amount of circulating tumour cells by their short-term expansion. By using this protocol, we overcame the stumbling block of the rare population by exploiting their intrinsic prerogative to proliferate.

The analysis of their secretome provides direct information about the amount and type of secreted molecules by cultured cells (14) and possibly on the activated metabolic pathways, as previously reported (12–14). In particular, we previously described the cellular landscape characterizing the BDCs from cancer and healthy subjects and demonstrated a proportional enrichment in circulating the recapitulating stage of cancer cells and the primary tumour proliferation degree [3].

Circulating tumour cells (CTCs) are cancer cells with a solid tumour origin recognized in the peripheral blood, their standard molecular definition is based on the positive detection of cells with an intact nucleus expressing cytokeratin (CK) but negative for CD45, a white blood cell (WBC) marker [4]. Moreover, we demonstrated that the secretome produced by cancer BDCs were superimposable to the intratumoral interstitial fluid extract from the primary lesion in the same patient for cytokines and glycan adducts content [5,6,7]. In this way, we analyzed here the secretome of BDCs focusing on the cytokines production and the advanced glycation products [7]. The analysis of the secretome of blood-derived cultures from cancer patients at different stages of the disease, provides a cross-section of knowledge of the molecules that alternate in production and secretion during the progressive steps of the disease. Hence, using an appropriate approach for the analysis of secretome from cultured circulating cancer cells, represents a crucial tool to discover new biomarkers. Here, we analyzed and correlated the content of adducts of methylglyoxal referencing the oxidative stress and secreted cytokines as an inflammation reference in the secretome obtained from blood-derived cultures.

It is well known from the early past century that cancer cells switch their metabolic profile enhancing the glucose uptake so, today this characteristic is exploited in cancer imaging. This feature is very often associated with an increased glycolytic rate that is maintained even in conditions of high oxygen pressure and is named the Warburg phenomenon by its discoverer [8].

In addition to the intracellular high glucose concentration itself, a high rate of glycolysis produces, in a non-enzymatic way, a certain number of aldehydes and ketones as intermediate products, that could react with amino groups of proteins and participate in the generation of advanced glycation end products (AGEs). Methylglyoxal (MG), among them, is considered the major precursor of AGEs or advanced glycation adducts (AGAs) for its ability to irreversibly modify proteins under physiological conditions [9].

Advanced glycation adducts (AGAs) accumulate in tissues during several pathological processes that include diabetes and cancer and can modulate the cell function by binding with multiple cell surface binding sites among which the best characterized is the signal transduction receptor known as RAGE (receptor for advanced glycation end-products). RAGE is a pattern-recognition receptor recognizing endogenous molecules, such as AGAs, released in the context of infection, physiological stress, or chronic inflammation [10,11]. Binding of the AGA with its receptors results in multiple effects including the release of pro-inflammatory cytokines and oxidative stress in multiple pathologies such as diabetes, heart dysfunction, and cancer [5,7]. A recent analysis of BDCs secretomes obtained from cancer patients, showed concentrations of methylglyoxal adducts (MGAs) that correlated with the stage of disease and the tumour proliferation degree [3].

In the present study, we analyzed by using liquid biopsy and short-term expansion, how cancer cells, from diverse stages of the illness, are able to condition the tumour microenvironment by virtue of their altered metabolism. Specifically, we tried to dissect the relationship between glycation adducts richness, which is the expression of metabolic switch, and the specific pattern of secreted cytokines inside their own secretome, with the prospect of understanding how this molecular signature could influence the tumour microenvironment, the diagnostic and prognostic contexts.

## 2. Results

### 2.1. Secretome Characterization and Oxidation Profile

Expanded cells were analyzed for CD45 expression to evaluate the non-hematological cellular compartment within BDCs. In BDCs from a healthy subject the mean percentage (mean ± standard deviation, SD) of CD45^+^ was 94 ± 4, and CD45^−^ was 6 ± 4. In the patient’s group, it was 37 ± 12 CD45^+^ and 63 ± 12 CD45^−^ (Figure 1). A comparative analysis between the two clinical groups showed a highly significant difference of *p* = 0.0014 for prevalent CD45^−^ in the patient’s group. Moreover, within the patient’s group, CD45^−^ cells increased in patients with a high histological grade (*p* < 0.0001). Moreover, the non-hematological population from cancer patients was evaluated for the expression of Pan-CK showing a high prevalence of these cells in BDCs of advanced cancer cases (*p* = 0.002).

After two weeks, 800 µL of a culture medium was collected from all prepared BDCs. The secretome was observed at an atomic force microscopy (AFM) highlighting a different composition between cancer and healthy subjects, as previously demonstrated [7]. In particular, results from the AFM analysis showed that healthy subjects and individuals with inflammation were characterized by the prevalence of secretive vesicles of a minor size with respect to those found in the secretome of cancer patients, as shown in Figure 1. These morphological features are suggestive of a different complexity of the tumour secretive pattern as previously demonstrated both in tumour tissues [12] and in the cancer secretome of liquid biopsy [7]. Malara et al. demonstrated a different content of DNA, RNA, proteins in the secretome of blood-derived cultures, and a high correlation with the interstitial liquid taken from the tumour tissue in some cancer cases [7].

The secretomes were analyzed for their content of methylglyoxal adducts by the immunoblotting assay. The mean value was 0.5 (± 0,09) and 0.8 (± 0,05) au for healthy and cancer samples, respectively (Figure 2a). A statistically significant difference between the two cohorts of samples was found (*p* = 0.01). These results confirmed the compromised oxidation profile of the patients compared to the healthy subjects. These data were sustained by the immunofluorescence assay carried out by using an antibody against DJ-1. The multifunctional deglycase protein DJ-1 also brings antioxidative properties. This protein is mainly able to repair methylglyoxal- and glyoxal-glycated adducted proteins, releasing cleaned proteins and lactate or glycolate, respectively [13]. DJ-1 prevents the formation of advanced glycation end-products (AGE) that cause irreversible damage activating on early glycation intermediates (hemithioacetals and aminocarbinols) [14,15]. Short-term expanded cells within blood-derived cultures were analyzed for the expression of DJ-1 (green fluorescence in Figure 2b) expressed in 68% of the cancer cells and in 20% of healthy cells with a significate difference between the two samples (*p* = 0.01). Figure 2b reports the DG-1 (green fluorescence) and MGAs (red fluorescence) immunoreactivity of short-term expanded circulating tumour cells in a case of glioblastoma. In the literature, the overexpression of DG-1 in glioblastoma has prognostic implications [16].

### 2.2. Secreted Cytokines Expression and Inflammation

The secretomes were assayed for the content of the following cytokines: IL-1α, IL-1β, IL-2, IL-4, IL-6, IL-8, IL-10, IFNα, TNFα, and VEGF. The hierarchical clustering based on Ward’s method for each interleukin, in Figure 2c, showed a distribution of the cytokines in two major clusters corresponding to healthy subjects (HS) and localized cancer (KL) samples in respect to the advanced cancer patients (KA), suggesting different patterns of inflammation between these two cohorts of samples. The comparative evaluations for each interleukin, displayed in Figure 2c, show the difference observed in the secretome of cancer in respect to the progression stage of the disease. Within the cancer cohort, considering two classes of samples in order to the stage of disease, a significant increase of the IL-10 secretion in advanced cancer cases was observed, as shown in Figure 2c, as well as an increment of IL-1β, IL-2, IL-8, in the localized cases. No significant differences were observed for the secretion of other cytokines.

### 2.3. Correlation Between Oxidation Profile and Inflammation Patterns

The analysis of correlation performed between the content of MGAs and specific interleukins secretion was performed to evaluate if the quality of cytokines production could be influenced by the progressive altered metabolism and consequent comprising oxidative balance registered with the progression of the disease. A negative correlation between MGAs and the secretion of some specific interleukins in the localized cancer cases, reverting in a positive correlation in the advanced cancer cases, was observed and reported in Figure 3a,b. Moreover, in the healthy secretome, a positive correlation was found in 10% of healthy samples between IL-10 and IL-1β. A negative correlation was found for the other cytokine investigated.

## 3. Discussion

Our liquid biopsy workflow (Malara BDCs protocol), favours survival and expansion of circulating cancer cells reducing the hematological quota by means of a specific gradient. Successively, our culture conditions promote the growth of non-hematological cells and employ a standardized culture medium. The efficacy of this is the same protocol that has been demonstrated in healthy subjects for the preferential expansion of non-hematological cellular elements (for example endothelial cells) expandable to a minor extent in respect to the cancerous elements in BDCs [17,18,19,20,21]. The prevalent type of expanded cells in the BDCs conditioned the composition of a relative secretome, providing data about the relative microenvironment. The investigation on the cytokines and MAGs present in the BDCs secretome from healthy subjects and cancer patients, is based on this logic. Methylglyoxal (MG) protein modification is not specific: Methylglyoxal targets the amino groups present in lysine, arginine, and cysteine side chains of all proteins (also N-terminal amino groups of other macromolecules such as phospholipids and nucleic acids) by the Maillard reaction or glycation. The antibody we used is raisen against an MG-modified ovalbumin, so it is able to recognize these protein adducts. The aim of the immunoblot analysis was to prove, in this setting of collected secretomes, that there is a correlation between the stage of cancer disease, the concentration of MGAs (methylglyoxal adducts), and the panel of produced cytokines in the secretome.

The prevalent cytokines secreted by tumour cells in the different stages of disease are reported in the cytokine cluster panel in Figure 2. In particular, in the secretome of BDCs of subjects with advanced tumours, IL-10, VEGF, and IL-1 beta are prevalently present. This pattern appears correspondent with the known biological conditions typically induced by metastatic cancer cells as the immunosuppressive function (IL-10), vasculogenic action (VEGF), and hypoxic condition (IL-1b) [22]. 

In these cohort of secretomes a positive correlation between the production of IL-6, IL-1β, IL-2, and IL-4 and the concentration of MGAs has been found. On the other hand, the IL-8 appears to be produced in patients with the localized tumour and released in a correspondent secretome, in the function of MGAs abundancy. IL-8 has an inductive function on the mesenchymal epithelial transformation (EMT). The presence of this cytokine in this cancer phase suggests a potential intriguing influence between the increase in MGAs levels in the EMT. On this way, recently, a strong influence of the cancer cell metabolism to control EMT progression and induce tumor aggressiveness has been reported [23].

However, in Figure 2c, the localized tumour was clustered prevalently with healthy subjects. In this cluster, the prevalent cytokines produced were IL-6 and IL-8, INFγ, TNFα, and IL-1β. This cytokines pattern presents a heterogeneous composition in agreement with the heterogeneous clinical spectrum presented by healthy subjects with and without inflammation and localized cancer patients grouped here.

Moreover, as mentioned above [11,24,25], the production of certain cytokines is strictly dependent on the type of cellular metabolism. In cells with a glycolytic metabolism, an increased collateral production of aldehydes such as methylglyoxal and glyoxal is present. The methylglyoxal and glyoxal complex with the proteins, and eliminate amino groups then change their protonation state [7] when forming glycated adducts; this could be responsible for a compromised DJ-1 function, as suggested by the altered DJ-1 expression shown in Figure 2b. MAGs interact with the RAGE receptor favoring the activation of NF-kB and inducing the secretion of a specific cytokine pattern. In the secretome of patients with advanced carcinoma, it has been possible to find a positive correlation between the secretion of IL-6, IL-1β, IL-2, and IL-4 and the concentration of MGAs. These data support the concept that the degree of metabolic switching in cancer cells increases with the disease progression in addition to the typical pattern of produced cytokines. Moreover, the lower concentrations of the glycation products characterizing the localized tumour phase, correlate less with the production of cytokines suggesting that the balance between these two systems is very delicate and precarious. In this regard, our results suggest that the presence of regulating mechanisms associated with the metabolic set, act through a gene-specific regulation or pharmacological cytokine inhibition. This could teach us that promising treatment strategies may exploit these same interconnections and could be applied to prevent tumour progression to improve antitumour therapeutic efficacy.

When we correlated MGAs and cytokines production in BDCs secretome of healthy subjects, we distinguished a group of subjects with a basal production of cytokines that did not correlate with the MGAs content in a corresponding secretome sample. Moreover, a second group was observed to be characterized by relative high levels of IL-10 and IL-1β in their secretomes. Both these cytokines have a negative immunomodulatory role. In particular, IL-1β that is generated within the tumour microenvironment predominantly by tumour-infiltrating macrophages, can promote tumour growth and metastasis via different mechanisms [26]. IL-10 is an immunomodulatory cytokine, upregulated in various types of cancer; its correlation with the disease progression indicates a critical role in the tumour microenvironment [27].

In Figure 3b, we summarize the main information obtained by analyzing the secretome of primary cultures, enriched in circulating tumour cells by short-term expansion. In healthy subjects, the immunosuppressive IL-1b and IL-10 were the prevalent cytokines detected in the microenvironment. Moreover, a prevalent level of IL-8, involved in the EMT, correspond to the progressive increase of glycated adducts production and their release in the secretome. When the content of glycated adducts reaches the highest level, the panel of cytokines is heterogenous reflecting the heterogeneous composition of tumour clonal expansion.

These results provide an important first step towards improving an integrated metabolomics and cytokine profiling approach as a potential to aid in the identification of important implications for cancer monitoring and long-term survival. The metabolomic analyses of fluids and tissues from cancer patients improve our knowledge of the reprogramming of metabolic pathways involved; also, these same pathways enable us to understand the pathophysiologic mechanisms that underlie the resistance to chemotherapy in treated patients [28]. In conclusion, our study highlights a personalized and non-invasive observation point by using the BDCs secretome characterization, opening up the opportunity for detection and capturing dynamic changes of cancer in real time for the integration of metabolomics and cytokine profiling approach.

## 4. Materials and Methods

### 4.1. Experimental Model and Subjects Details

Prospectively, blood samples have been collected from July 2013 to the present day, from patients that arrived at the Clinical and Experimental Medicine and Medical and Surgical Sciences Departments of University “Magna Græcia” of Catanzaro. The study named CHARACTEX (CHARActerization of Circulating Tumour cells and EXpansion) was approved, under number 2013.34. An informed consent was obtained by all enrolled subjects. Peripheral blood samples (total volume of 5 mL) were drawn from the volunteer subjects and placed into tubes containing EDTA as an anticoagulant. All subjects enrolled in this study, had ‘normal’ glucose level of 70 to 85 mg/dL and a family history without cases of diabetes and/or neurodegenerative diseases. Subsequently, the samples were processed following the procedure described previously by Malara et al. [7,29]. The enrolled volunteers’ details are resumed in Table 1.

### 4.2. Primary Blood Derived Cultures and Characterization

The cell fraction of interest was taken from a gradient phase previously defined as a working range [7,29]. After washing in 1 X phosphate buffered saline (PBS) the cells were collected and expanded for 14 days (short-term cultivation). The mean of the cell collected and seeded in Petri dishes was 3 (0.7) × 10^5^ and 2.7 (0.9) × 10^5^, respectively for cancer and healthy volunteers. The medium composition employed in the expansion phase was reported by Malara et al. [29,30]. Cultured cells were stained with hematoxylin and eosin (H&E) staining. Moreover, cultured cells were analyzed with anti-CD45 and anti-Pan-CK antibodies to identify different populations of naïve hematological and non-hematological cells. Instrument performance and data reproducibility were monitored using a Cytometer Setup & Tracking Module (BD) and checked through the acquisition of Rainbow Beads (BD). Compensations were evaluated using CompBeads (BD). Flow cytometry was performed with a FACS ARIA III (Becton Dickinson) analyzed with FACSDiva v. 6.1.3 and FACSuite v.1.05 (BD) software.

For cellular immunofluorescent staining, cells on slides were fixed with 4% paraformaldehyde for 15 min at 22 °C. Then, the samples were washed three times with 1 X PBS and then blocked with PBS containing 0.2% Triton X-100 and a 10% bovine serum albumin for 1 h. Subsequently, the samples were then incubated with a primary antibody (1:1000) for 16 h at 4 °C and washed three times with PBS. Next, cells were incubated with an Alexa 488-conjugated secondary antibody for 3 h at 22 °C and after DAPI staining were viewed by using a laser confocal microscope (Nikon TI-E) (antibodies list is reported in Table 2). The TI-E microscope software was used to morphometrically analyze the fluorescence intensity.

### 4.3. Secretome Collection and Characterization

The cell cultures were monitored at 48 h intervals and each time 10% of the total volume of the culture medium was collected and replaced with a fresh medium. The 10% of the collected medium was placed in a cuvette and stored at 4 °C. After two-week incubation, the cell culture was harvested, and the medium was separated from the cellular elements by centrifugation at 1870 rpm for 15 min. The supernatant was added to the previous collected medium, filtered, and stored at -80 °C. The pellet formed was collected and used for successive characterizations.

### 4.4. Cytokines and Growth Factors Array

Cytokine and growth factor levels were evaluated simultaneously using the “Cytokine & Growth Factors Array (CTK)” kit, the Evidence Investigator biochip analyzer (Randox Labs, UK), and by the Proteome Profiler kit (R&D system).

### 4.5. Immunoblotting

After total proteins purification, the sample concentration was assayed by using the Bradford Protein Assay (Bio-Rad). Fifty micrograms from each protein sample were loaded onto a 12% SDS-polyacrylamide gel and electrophoresed at 80 V. After separation, proteins were transferred onto a nitrocellulose membrane by the Trans-Blot Turbo (Bio-Rad) protein transfer system. Membranes were first blocked in 5% nonfat milk/1 X TBS-T and then incubated overnight at 4 °C, with a 1:500 dilution in the same blocking buffer of mouse anti-methylglyoxal-protein adducts (Cell Biolabs, Segrate, Italy). After washing, membranes were incubated with a 1:2000 dilution of HRP-conjugated anti-mouse IgG (Dako) (antibodies list is reported in Table 2). Immunocomplexes were detected by using the SuperSignal West Femto ECL substrate (Pierce, EuroClone). A fully automated densitometric software, Alliance 2.7 1D (UVITEC, Eppendorf, Milan, Italy) has been used for acquisition and analysis of immunoblots images [31,32]. The relative amount of MGAs in the cell proteasoma has been normalized against the total protein load for each lane, by calculating the intensity ratio with the corresponding lane, stained with a Ponceau S dye, and expressed as arbitrary units (au). The western blot analysis was replicated two times for each sample.

### 4.6. Atomic Force Microscopy Imaging and Analysis

Freshly cleaved mica disks (TedPella Inc., USA) were treated with three amino propyl-trimethoxysilane (APTMS) at 0.1% v/v in Milli Q water for 3 min, removing the silane excess with 3 mL of water. Before deposition, five secretome samples, belonging to three groups (healthy (A), thyroiditis (B), and in thyroid carcinoma (C)) were concentrated three times, by evaporating 100 µL of the sample for 24 h at 25 °C, and were then incubated for 60 min on positively charged mica disks obtained as described above. Just before imaging, 70 µL of Milli Q water were added to the samples. Images have been acquired using an Oxford Instrument Cypher instrument, equipped with an Environmental Scanner. All acquisitions have been obtained by scanning in liquid at a temperature of 20 °C, using BL-AC40TS probes (Olympus Corporation, Japan) with a nominal resonance frequency in air of 110 kHz. At least four 15 × 15 µm^2^ scans for every sample has been acquired for statistical analysis. Sample details have been acquired as scanning areas of ~300 × 300 nm^2^. Data have been imported in ImageJ for imaging and data analysis. Particles were detected in ImageJ by thresholding (height > 10 nm, area > 100 nm^2^), and their normalized height distribution has been plotted using a bin width of 5 nm.

### 4.7. Statistical Analysis

Comparison between patients and the control group was performed using Mann-Whitney and Kolmogorov-Smirnov tests with a valid statistical significance of *p* < 0.05. Sub-groups were compared using the T-test (for continuous variables) and Chi-square test or Fisher test (for categorical variables). Pearson’s correlation coefficient r with a p-value was performed for the correlation study. All statistical analyses were performed using MedCalc for Windows, version 18 (MedCalc Software, MariaKerke, Belgium). For hierarchical clustering, each interlukin was calculated using the correlation coefficient and Euclidean distance. Dendrograms were generated using Ward’s method. The authors presented the quantitative data as the mean and standard deviation (SD).

## Figures and Tables

**Figure 1 ijms-21-04711-f001:**
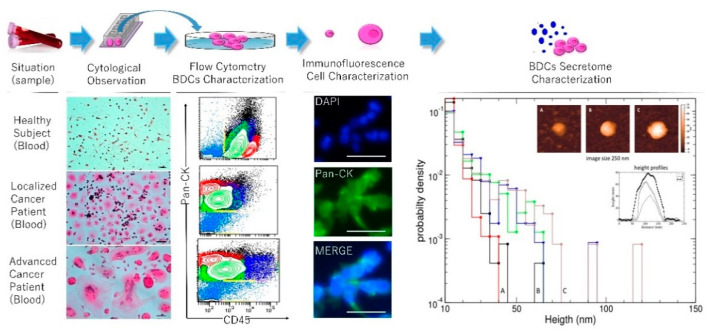
Blood-derived cultures and secretome characterization. The top panel describes an experimental workflow starting from the blood sampling up to the secretome characterization. H&E stained cytological preparations, were prepared from the blood-derived cell cultures (BDCs) derived from samples collected by healthy volunteers and cancer patients in different stages of two reported cases of breast cancer. The cultured cell population was analyzed for the expression of CD45 to evaluate the in vitro hematological (CD45^+^) and non-hematological (CD45^−^; Pan-CK^+^) compartment of the cells isolated and expanded. Immunofluorescence for the expression of Pan-CK in an advanced case of breast cancer was reported. The secretome complexity pattern was analyzed with AFM. Secretive vesicles (ranging from 50 to 250 nm in size) in healthy, thyroiditis, and in thyroid carcinoma blood-derived cultures secretome. The distribution density of particles height is plotted with a bin width of 5 nm. The insets show representative images of particles of each population, alongside with height profiles traced through the particles. Scale bars in cytological and immunofluorescence panels correspond to 50 µm.

**Figure 2 ijms-21-04711-f002:**
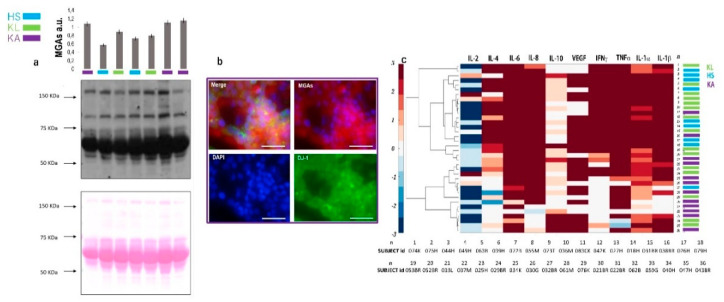
Secretome analysis. The top of the panel shows the histogram of the intensity rate with an error bar for each case analyzed for methylglyoxal adducts (MGAs) content by western blotting. In (**a**) the relative abundancy levels of MGAs in the secretomes of BDCs of healthy subjects, (HS in blue), localized (KL in green), and advanced cancer (KA in violet) cases. The relative amount of MGAs in the cell proteasoma has been normalized against the total protein load for each lane, stained with a Ponceau S dye and expressed as arbitrary units (au). (**b**) Immunofluorescence DJ-1 and MGAs in BDCs. Immunoreactivity is reported by a fluorescence light for DJ-1 (green) and for MGAs (red) in short-term BDCs from glioblastoma (required in violet), characterized by a prevalence of circulating tumour cells co-expressing DJ-1 and MGAs. Nuclei counterstain appears in blue fluorescence for DAPI incubation. (**c**) Hierarchical clustering based on Ward’s method grouped healthy subjects (HS) and patients with localized (KL) and advanced (KA) cancer in the function of each interleukins produced and presented in the correspondent BDCs secretome samples. Scale bars in panel (**b**) correspond to 50 µm.

**Figure 3 ijms-21-04711-f003:**
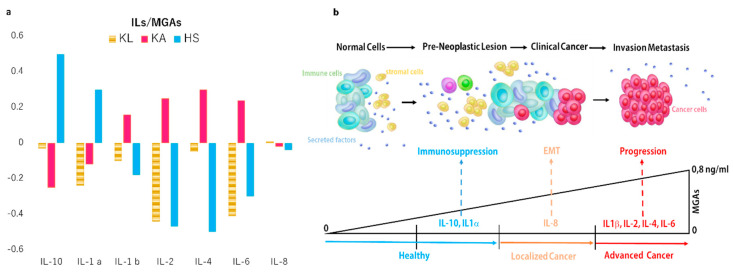
Interplay between metabolism and cytokines production. (**a**) Graphic representations of interleukins patterns in the function of MGAs secretion was reported as coefficient correlation values for each secreted cytokine. Moreover, based on the correlation value observed in the reassuming graphic (**b**) suggests a potential interpretation. In particular, in a healthy group of secretome samples, two classes of subjects are evident as distinguished for the ILs-pattern of IL-10 and IL-1. The cancer group is distinguished in localized and advanced phases of the disease. In the secretome of localized cancer BDCs, IL-8 appears to be prevalently secreted in respect to other ILs. In the advanced cancer cases, the secretome of BDCs are prevalent to the secretion of IL-1β, IL-2, IL-4, and IL-6 as a consequence of the increased levels of MGAs inducing NFKB by the receptor for advanced glycation end-product (RAGE) way and relative interleukins cascade activation.

**Table 1 ijms-21-04711-t001:** Characteristics of volunteers involved.

Cancer Cases			Healthy Subjects		
Characteristi	Localized	Advanced	Characteristic	Without Pre-Cancerous Conditions	With Pre-Cancerous Conditions
Total	18	20	Total	12	2
Age mean (SD)	54 (16)	60 (16)	Age mean (SD)	54 (13)	60
Male (%)	6 (33)	4 (20)	Male	2 (16)	2 (100) *
Stage (%)	I (50) II (50)	III (65) IV (35)	Comorbidity	2 (16) **	1 (50) ***
Melanoma (%)	4 (22)	2 (10)	* Ulcerative recto colitis 1 + 1 multinodular thyroid gland ** Hypertension *** Atrial Fibrillation		
Breast cancer (%) *	9 (50)	12 (60)			
Colon cancer (%) **	4 (22)	2 (10)			
Thyroid cancer (%) ***	1 (5)	0 (0)			
Glioblastoma (%)	0 (0)	4 (20)			
* 70% Intraductal and 30 % lobular Breast cancer ** Colon Adenocarcinoma *** Medullary thyroid Cancer					

**Table 2 ijms-21-04711-t002:** Antibodies list.

Reagent or Resource	Source	Identifier
CD45	Clone D1 Becton and Dickinson	Cat#564327
Pan-CK	Clone C-11 Abcam	Cat#Ab 106166
Anti-methylglyoxal adducts	Biolabs	Cat#STA-011
Anti-mouse IgG	Thermofischer	Cat#A-21052
Polyclonal Rabbit anti human DJ-1 antibody	NovusBiolgocials	NB#100-483
Anti-rabbit IgG	Cell Signaling	Cat#A210070

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
