# Peer review of "Microenvironment Molecular Profile Combining Glycation Adducts and Cytokines Patterns on Secretome of Short-term Blood-derived Cultures during Tumour Progression"

_ijms, 2020, doi:10.3390/ijms21134711_

Round 1
Reviewer 1 Report
Microenvironment molecular profile combining glycation adducts and cytokine patterns on secretome of short term blood derived culture during tumor progression
Summary: The author compared secretome of blood derived cell from both healthy subjects and cancer patients at different disease status. Authors clarified most of comment in the revised manuscript. However, several comments need to be more explained.
- The authors should address the reason for examining CD45 in figure 1.
- labeling are missing in figure 1 and appropriate labeling need to be presented in results part.
- What is difference between the upper and lower panel in figure 1 and figure 2?
- IFC and heatmap data seems same in upper and lower panel in figure 2. Please clarify this issue
- which band represent MGAs in figure 2a (lower panel)?
Therefore, the current form of this manuscript is not suitable for international journal of molecular science.
Author Response
Manuscript ID: ijms-825388
Title: Microenvironment molecular profile combining glycation adducts and cytokines patterns on secretome of short-term blood-derived cultures during tumor progression
Journal: IJMS
Dear Editor,
we are pleased to submit to International Journal of Molecular Sciences, Special Issue on Cellular Secretomes, a revised version of the manuscript entitled Microenvironment molecular profile combining glycation adducts and cytokines patterns on secretome of short-term blood-derived cultures during tumor progression, and a detailed response to the comments of the Reviewers. The manuscript has been revised. Following suggestions from the Reviewers:
- We have rewritten parts of the manuscript
- We have included comment in the article related to the comment required by reviwers
- We have modified the figures and highlighted the bars
- We have included details on the primary culture blood-derived regarding the number of cells seeded
- We have discussed the biochemical analogies between the secretome of blood-derived culture with the Tumor interstitial fluid
Considering the corrections made to the manuscript we hope that the paper can be accepted for publication in this present form. We warmly thank the reviewers that, with their comments, contributed to improve the work.
In the following, you will find the original comments from the reviewers (bold black text) and the point-by-point response of the authors (bold blue text). In the manuscript all modified portions are tracked
With Regards,
the Authors
Reviewer 1
Microenvironment molecular profile combining glycation adducts and cytokine patterns on secretome of short term blood derived culture during tumor progression
Summary: The author compared secretome of blood derived cell from both healthy subjects and cancer patients at different disease status. Authors clarified most of comment in the revised manuscript. However, several comments need to be more explained.
- The authors should address the reason for examining CD45 in figure 1.
In agreement with the observation of the reviewer, we have highlighted in the text at paragraph 3.1 lines 171 and 172 the reason for examining CD45 expression. The circulating tumor cell population is numerically lower in proportion to the hematologic cells. The expression for CD45 is traditionally used in the liquid biopsy field, to distinguish the epithelial cells CD45 negative form the prevalent haematological cells expressing CD45 antigen on their surface.
- labeling are missing in figure 1 and appropriate labeling need to be presented in results part.
The description required has been conveniently included in the paper.
- What is difference between the upper and lower panel in figure 1 and figure 2?
The description required has been conveniently included in the paper.
- IFC and heatmap data seems same in upper and lower panel in figure 2. Please clarify this issue
The description required has been conveniently included in the paper.
- which band represent MGAs in figure 2a (lower panel)?
Methylglyoxal (MG) protein modification is not specific: methylglyoxal targets the amino groups present in lysine, arginine and cysteine side chains of all proteins (also N-terminal amino groups of other macromolecules like phospholipids and nucleic acids) by the Maillard reaction or glycation. The antibody we used is raisen against MG-modified ovalbumin, so it is able to recognize these protein adducts. The aim of our immunoblot analysis is to prove that there is a correlation between the stage of cancer disease and the concentration of MGAs (methylglyoxal adducts) in secretoma. We improved the description of this assay in material and methods section and in figure 2a caption, for better explain the logic of this analysis clarifying why we compared immunoblots with Ponceau S staining.

Reviewer 2 Report
The abstract and the text needs to be rewritten by a native speaker. It is not very clear. The study is not very clear as the authors state that the more the tumor progresses the more circulating cells are released. From material and method it is not clear how many cells were plated in which volume. It is well-known that the strong expansion of tumor cells induces the accumulation of mutations not present in the initial tumor cells due to the genetic instability of tumor cells (10.1038/s41568-018-0095-3). It is well known that the cell growth in 2D changes the cells (10.5114/aoms.2016.63743) and most probable also the secretome which serves for the cell-cell communication. Moreover, the circulating cancer cells are already modified to be released from the primary tumor and therefore no longer represent the features and secretome release profile of the original tumor. What about the distinction between the different tumor types. There are too many open questions. Also a more detailed characterization of the plated cells is missing
Line 46-48: The statement needs consideration. Circulating cancer cells are already modified to be released so I am not sure that from these cells and their cell culture conclusions can be drawn on the microenvironment of the original tumor. Moreover, the low concentration of the cells will lead to more mutations which are not present in the original tumor due to the numerous cell divisions to achieve sufficient cells for analysis.
Line 97: The method part lacks information. e.g. which cells were collected? How many cells were plated in which culture dish? etc.
Line 168-170: the amount of non-hematologic CD45 negative cells in tumor patients is too high which indicates that there is a preferential growth of CD45 negative cells or a faster expansion. However, the total number of cell in culture of healthy donors to tumor patients is not stated.
Literature of other authors regarding the role of the secretome and their content of biomarkers are missing (e.g. Oliveira et al. 2020 or Xue et al. 2008) while the authors included many self-citation even if they do not fit the context. 12 out of 30 citations are from one of the authors.
Error bars are missing. The number of independent experiments is missing. the writing in the figures is too small and therefore illegible.
Author Response
Manuscript ID: ijms-825388
Title: Microenvironment molecular profile combining glycation adducts and cytokines patterns on secretome of short-term blood-derived cultures during tumor progression
Journal: IJMS
Dear Editor,
we are pleased to submit to International Journal of Molecular Sciences, Special Issue on Cellular Secretomes, a revised version of the manuscript entitled Microenvironment molecular profile combining glycation adducts and cytokines patterns on secretome of short-term blood-derived cultures during tumor progression, and a detailed response to the comments of the Reviewers. The manuscript has been revised. Following suggestions from the Reviewers:
- We have rewritten parts of the manuscript
- We have included comment in the article related to the comment required by reviwers
- We have modified the figures and highlighted the bars
- We have included details on the primary culture blood-derived regarding the number of cells seeded
- We have discussed the biochemical analogies between the secretome of blood-derived culture with the Tumor interstitial fluid
Considering the corrections made to the manuscript we hope that the paper can be accepted for publication in this present form. We warmly thank the reviewers that, with their comments, contributed to improve the work.
In the following, you will find the original comments from the reviewers (bold black text) and the point-by-point response of the authors (bold blue text). In the manuscript all modified portions are tracked
With Regards,
the Authors
Reviewer 2
- The abstract and the text needs to be rewritten by a native speaker. It is not very clear.
We appreciate the observation of the reviewer and the text needs have been rewritten
- The study is not very clear as the authors state that the more the tumor progresses the more circulating cells are released.
The authors thank the reviewer for this clarification. It is necessary to specify that the data was reported in this article as one of the results described in paragraph 3.1, in figure 1 and in the discussion lines 323 to 333. The progressive increase in the number of cancer cells in patients with advanced cancer compared to patients with localized cancer it has also been demonstrated by our group in the previous publications mentioned in references 15 and 14 and in the work recently published on Cancers Coluccio et al, included in this new version of the manuscript
- From material and method it is not clear how many cells were plated in which volume.
The description required has been conveniently included in the paper in material and methods section
- It is well-known that the strong expansion of tumor cells induces the accumulation of mutations not present in the initial tumor cells due to the genetic instability of tumor cells (10.1038/s41568-018-0095-3).
It is well known that the cell growth in 2D changes the cells (10.5114/aoms.2016.63743) and most probable also the secretome which serves for the cell-cell communication.
The authors thank the reviewer for this clarification. It is necessary to specify that the despite the advances in sequencing technology and target mutation identification have had a major impact, the responses to targeted therapies among genetically defined patients are heterogeneous. Matching therapeutics to genetic mutations is currently limited by an incomplete understanding of the interplay between tumor cell and its microenviroment. Therefore, the developmet of primary patient-derived cancer cells using biopsy 2D cultures of mixed cell populations growing in defined media represent a functional diagnostic assay useful to match therapies to individual cancer patients.
Kodack DP, Farago AF, Dastur A, et al. Primary Patient-Derived Cancer Cells and Their Potential for Personalized Cancer Patient Care. Cell Rep. 2017;21(11):3298‐3309. doi:10.1016/j.celrep.2017.11.051
Over the last few years, circulating tumor cells have become the subject of researchers for their great potential as no-invasive diagnostic and prognostic biomarker also for their potential in providing functional and molecular data.
Danila, Daniel C et al. “Circulating tumors cells as biomarkers: progress toward biomarker qualification.” Cancer journal (Sudbury, Mass.) vol. 17,6 (2011): 438-50. doi:10.1097/PPO.0b013e31823e69ac
The main problem in their isolation is their limited availability. Many techniques have been developed and are continuously improving. In our laboratory, from 2008 an isolation protocol is used for increasing the availability of circulating tumor cells by their short-term expansion, only 14 days, as reported in Material and Method. By using this protocol, we passed the problem of the rare population by exploiting their intrinsic prerogative to proliferate.
This description has been conveniently included in the paper in discussion section.
- Moreover, the circulating cancer cells are already modified to be released from the primary tumor and therefore no longer represent the features and secretome release profile of the original tumor.
The authors thank the reviewer for this clarification. It is necessary to specify that the production of cultures derived from the periphery blood samples from cancer patients, similar to the tumor biopsy derived primary culture cells, offers the possibility of having viable tumor cells, representing a good model to study. Therefore, the analysis of their secretome provide direct information on the quantity and type of secreted molecules by cultured cells (14) and on the actives metabolic pathways, as previously reported (12-14). In particular, we previously described the cellular landscape characterizing the BDCs from cancer and healthy subjects demonstrating a proportional enrichment for circulating cancer cells to the stage and the proliferation degree of the primary tumor as published on Cancers Coluccio et al, included in this new version of the manuscript
Circulating tumor cells (CTCs) are cancer cells of solid tumor origin recognized in the peripheral blood, their standard molecular definition is based on positive detection of cells with an intact nucleus expressing cytokeratin (CK) but negative for CD45, a white blood cell (WBC) marker.
Allard WJ, Matera J, Miller MC, Repollet M, Connelly MC, Rao C, et al. Tumor cells circulate in the peripheral blood of all major carcinomas but not in healthy subjects or patients with nonmalignant diseases. Clin Cancer Res. 2004;10(20):6897–904. 10.1158/1078-0432.CCR-04-0378
Moreover, we demonstrated that the secretoma produced by cancer BDCs were superimposable to the intratumoral interstitial fluid extract from the primary lesion in the same patient for cytokines and glycan adducts content (12-14)
This description has been conveniently included in the paper in introduction section.
- What about the distinction between the different tumor types.
The description required has been conveniently included in the paper in the table 1 in material and method section.
- There are too many open questions. Also a more detailed characterization of the plated cells is missing
The description required has been conveniently included in the paper in material and methods section.
- Line 46-48: The statement needs consideration. Circulating cancer cells are already modified to be released so I am not sure that from these cells and their cell culture conclusions can be drawn on the microenvironment of the original tumor.
Please see the answer of the authors at point 5
- Moreover, the low concentration of the cells will lead to more mutations which are not present in the original tumor due to the numerous cell divisions to achieve sufficient cells for analysis.
Please see the answer of the authors at point 5
- Line 97: The method part lacks information. e.g. which cells were collected? How many cells were plated in which culture dish? etc.
This description has been conveniently included in the paper.
- Line 168-170: the amount of non-hematologic CD45 negative cells in tumor patients is too high which indicates that there is a preferential growth of CD45 negative cells or a faster expansion.
The authors thank the reviewer for this clarification. It is necessary to specify that the cells CD45 negative in BDCs form cancer patients correspond to the circulating tumor cells and their growth is in function to their intrinsic feature that is to proliferate more and better respect to untransformed cells. The circulating tumor cells isolated in this study were detected by the negative expression for CD45 and the positive expression for pan-CK as reported in figure 1 and in paragraph 3.1, results section.
- However, the total number of cell in culture of healthy donors to tumor patients is not stated.
This description is included in the paper in material and method section.
- Literature of other authors regarding the role of the secretome and their content of biomarkers are missing (e.g. Oliveira et al. 2020 or Xue et al. 2008) while the authors included many self-citation even if they do not fit the context.
The authors, in agreement with the reviewer, have inserted the indicated references. On the other hand, it is necessary to point out to the auditor that the works cited are necessary to support a protocol that originated by our group and it is therefore natural to report previous work already published in support of the new data presented here.
12 out of 30 citations are from one of the authors.
As reported above, the short-term culture protocol from the blood for the detection of circulating cancer cells exploiting their proliferative prerogative is a protocol developed and tested in the translational context for the moment only by our group.
- Error bars are missing.
This description has been conveniently made more evident in all figures.
- The number of independent experiments is missing.
The number of independent experiments is reported in the original manuscript in material and method section.
- the writing in the figures is too small and therefore illegible.
All the figures have been replaced with a more legible script

Round 2
Reviewer 1 Report
Authors answered all issue that I raised. Therefore, this revised manuscript is acceptable in International journal of molecular science.
This manuscript is a resubmission of an earlier submission. The following is a list of the peer review reports and author responses from that submission.
Round 1
Reviewer 1 Report
In the present manuscript Coluccio et al. present a characterization of secreted protein from blood cell cultures from diverse cancer patient as well as healthy individuals. Unfortunately, the experimental procedures do not get sufficiently clear, because the authors just refer to their previous work. Importantly, it is difficult to evaluate which populations of blood cells were exactly studied (which is finally mentioned in the discussion). The authors should present more information on these topics and restructure the manuscript, i.e. present experimental procedures and information on the scientific background in the respective sections, introduction and materials and methods.
The main outcome of the manuscript is the observation that the blood cells itself as well as their secretomes differ greatly between healthy individuals and cancer patients. This is not very surprising, however the present description is useful and the authors use various technologies to proof their observations. The authors also present data on different stages of cancer, which to me seems much more interesting and relevant in this context. If the necessary data is available, the authors might emphasize these analyses to improve the relevance of the manuscript and further consolidate their final conclusions presented in Figure 5.
For future studies, I recommend using diseased controls (underlying diseases, which are risk factors for development of the respective kinds of cancer) instead of healthy subjects. In such a, experimental set up, observed differences might finally proof to be really useful as biomarkers.
I have some specific comments as well:
The abbreviation MGA is not introduced in the text until the discussion section, where it is written MAG. Therefore, when reading the manuscript it remains unclear what exactly is represented by MGA in the figures. This structure is counterintuitive and frustrating!
Sufficient information on the exact types of cancers is missing for the investigated patients (supplementary table S2). This is highly relevant and of great interest and needs to be provided!
Importantly, no sufficient information on the used primary antibodies is provided and must be added in a revised version of the manuscript!
Regarding the methods the authors reference their previous publications, which is legitimate. However, it would be very useful and simplify the understanding of the experiments if the authors would provide a brief outline of the applied methods in the present manuscript. Unfortunately, it is not possible to understand what has been done without reading the earlier publications.
Regarding Figure 3, I cannot really recapitulate the clustering described by the authors. The labels on the right side of the figure represent the patients and experimental groups, but considering the labels no clustering is evident to me.
In the legend of Figure 4 the authors should include information on the applied statistical tests. It does not seem convincing to me that all comparison were significant with p<0.001. This is especially the case for IL1 alpha in the upper panel, where the means seem to be equal and the Standard deviation is huge (is the data really presented as mean +/- SD? This information is also missing!).
The presented data in Table 1 does not seem conclusive to me. The authors might present the data as scatter plots in addition, if necessary in the supplement. Most of the presented values are far away from suggesting a good correlation (if I understand Pearson correlation correctly, a perfect correlation would be r = 1). The presented data more suggests, that there is hardly any correlation between the individual cytokines. I cannot see how the reader should draw any useful conclusions from this. Moreover, I do not understand how these correlations support the authors final hypothesis illustrated in Figure 5.
The first paragraph of the discussion is a better introduction and contains much information which would be necessary to understand the manuscript from the beginning. The structure of the manuscript needs to be changed in this regard.
The authors derive a hypothesis (Figure 5) which is not completely supported by the clinical reality. Many types of cancer do not arise out of the blue, but are highly related to respective underlying diseases. However, instead of mentioning disease progression, the authors speculate about intermediate healthy subjects with a damaged oxidative stress repair system. As a matter of fact, oxidative stress can also be linked to many diseases which are also highly correlated to the development of cancer (e.g. in the liver). I suggest that the authors include some comment on cancer progression in the discussion and also refer to this in their figure.
Reviewer 2 Report
Title: Personalizing cancer risk profile combining glycation adducts and cytokines patterns on secretome of short-term blood-derived cultures
Summary: The authors compared the secretome of blood derived cell from both healthy subjects and cancer patients at different disease status to define a personalized oxidative profiling of cancer risk. However, the results are not enough to support the author’s suggestion without proper rationale.
Comments
- The author should clarify the meaning of oxidative status as well as MGAs. In figure 2 and table 1, they measured MGAs without any clear description.
- I think western blotting data missing. Even though information about immunoblotting was described in both material and method section and result (line 164-166), comparable results was not presented.
- The rationale and proper reason for analyzing DJ1 is not enough. Furthermore, direct correlation between DJ1 and secreted proteins such as interleukin family was not well described with proper results.
- Hierarchical clustering for interleukin family is hard to interpretate.
- The author presented graphical representations to supporting their suggestion. However, the results presented in this article are less suitable and not enough to support graphical representation.